# Accessing Occupational Health Services in the Southern African Development Community Region

**DOI:** 10.3390/ijerph17186767

**Published:** 2020-09-17

**Authors:** Masilu Daniel Masekameni, Dingani Moyo, Norman Khoza, Chimwemwe Chamdimba

**Affiliations:** 1Occupational Health Division, School of Public Health, University of the Witwatersrand, Parktown, Johannesburg 2193, South Africa; moyod@iwayafrica.co.zw (D.M.); NormanK@nepad.org (N.K.); 2Faculty of Social Sciences, Midlands State University, Gweru 9055, Zimbabwe; 3Baines Occupational Health Services and Safety Group, Harare 1410, Zimbabwe; 4Health Division, Programme Delivery and Coordination Directorate, African Union Development Agency-New Partnership for Africa’s Development (AUDA-NEPAD), Pretoria 0001, South Africa; Chimwemwe.Chamdimba@nepad.org

**Keywords:** occupational health services, mining, primary health clinics, labour

## Abstract

Only 15% of the global population has access to occupational safety and health services. In Africa, only 5% of employees working from major establishments have access to occupational health services (OHS). Access to primary health care (PHC) services is addressed in many settings and inclusion of OHS in these facilities might increase efficiency in preventing occupational diseases. A cross-sectional study was conducted in four Southern African Development Community (SADC) countries aiming at assessing the availability of OHS at PHC facilities and the organization of OHS. We conducted a literature review to assess the provision and organization of OHS services. In addition to the review, a total of 23 doctors from Zambia were interviewed using questionnaires in order to determine the availability of OHS and training. Consultations with heads of ministries were done in four SADC countries. Results showed that in the SADC region, OHS are fragmented and lack a comprehensive approach. In addition, out of 23 PHC facilities, only two (13%) provided occupational health and PHC. However, OHS provided at PHC facilities were limited to TB screening and audiometric testing. Our study showed a huge inadequacy of trained occupational health practitioners. This study supports the World Health Organization’s advocacy to integrate OHS at the PHC level.

## 1. Introduction

Recently, the International Labour Organization (ILO) indicated that about 2.8 million men and women die annually due to work-related problems [1]. Furthermore, about 86% of the total mortalities are attributed to occupational-health-related illnesses [2,3]. In the last two decades, industrial activities have increased, leading to the introduction of new hazards and new health outcomes [4]. The consequences of new hazards are likely to be more catastrophic in developing countries due to weaker technological advancement in occupational hygiene monitoring and occupational diseases diagnosis [1,5,6]. This suggests the need to develop mechanisms for practitioners in developing countries to advance their knowledge and understanding of the basic occupational health service package.

In the South African Development Community (SADC) region, the majority of people are employed in the mining sector [7]. Studies suggest that about half of the world’s vanadium, platinum, and diamonds originate in the region, while 36% of gold and 20% of cobalt are produced in this region [8]. Since mining operations have been associated with increased exposure to pollutants, leading to several health effects, the SADC region is considered amongst the top high-risk countries [9]. Exposure to fine mine dust has increased the risk of illnesses such as silicosis, coal pneumoconiosis, cardiovascular illness and tuberculosis [8,10,11].

Access to occupational health services and primary health care services can be one of the most appropriate ways of reducing associated health effects [7]. Though primary health care services are generally understood as a mandate of government to ensure universal health access mostly for the poor, this understanding needs to change [5,9]. OHS are often funded by industries but regulated by the government [12]. The understanding of both primary and occupational health services suggests that workers are not part of the general public nor are general public members’ part of the working population.

In many developing countries, there is an increasing challenge to access and provide occupational health services (OHS), as demand is on the rise due to employee’s awareness programs [6,9]. In the SADC region, the provision of OHS is poorly regulated [10,13]. Furthermore, there is a shortage of trained and knowledgeable occupational health practitioners [14]. Therefore, it is imperative to develop other mechanisms to increase capacity in OHS. Such alternatives might include the allocation of resources for training practitioners, establishment or review of legislative frameworks and adding OHS at primary health care centers [1,15].

The connection between labour and health is anticipated as a vehicle to integrate primary health care and OHS [3]. Most current- and ex-mine-workers are in communities where access to occupational health services is poor, but with good access to primary health care services, which are widely distributed [8]. This presents a good opportunity to integrate OHS into the primary health care system. Therefore, the introduction of OHS in primary health care is envisaged as a vital tool for accelerating the diagnosis and treatment of occupational diseases for those who are affected [3,11].

Resolution World Health Assembly (WHA) 60.26 by the World Health Organization (WHO) “Workers’ Health: Global Plan of Action”, calls for all member states to ensure coverage of all workers in both the formal and informal sectors across all working spaces, with essential interventions and basic occupational health services for the primary prevention of occupational and work-related diseases and injuries [16]. It is therefore imperative that occupational health services are provided at primary health care levels.

Several gaps in the organization of OHS in SADC regions are highlighted in several studies, which include lack of access to OHS, lack of training of health professionals on occupational health and poor regulatory frameworks. A case-based approach to the strengthening of OHS is required. In order to develop and implement an effective occupational health system, efforts are needed in the three spheres of effectiveness. The three spheres must encompass the following: technology (talking about training approaches and needs), economic (affordable health services) and societal (people’s perceptions, access and acceptance). Our study sought to evaluate the organization of OHS and determine the opportunities for integrating OHS into primary health care centers in four SADC countries: Zambia, Malawi, Mozambique, and Lesotho. The objective of this study was to determine the need for future integration of OHS into the primary health care institutions in the SADC region. The study hypothesizes that there is limited investment in OHS in both primary health care and occupational health facilities in the SADC region, despite high economic growth, especially in the mining sector.

## 2. Material and Methods

### 2.1. Study Area, Study Population and Data Collection

The study areas consisted of four SADC countries: Lesotho, Malawi, Mozambique and Zambia. The study adopted a mixed methods approach in collecting data, whereby a cross-sectional study, literature review and consultations of ministry heads were carried out to acquire sufficient information to gain an insight into the arrangement and provision of OHS in the SADC region. Research articles and technical reports from the four countries were reviewed and analysed. Several databases, which included Medscape/Medline, NIOSH, Science direct and the Social Science Citation Index, were used to access journal articles. The literature review process was Supplementary Materials by surveys and completion of databases using customised templates, on OHS arrangements. Through the project host, African Union Development Agency (AUDA-NEPAD), consultations on OHS were made with several heads of ministries in the four project countries. We undertook a direct search of relevant journals in the fields of occupational medicine/health, industrial hygiene, environmental health and public health. We also checked the citation lists of the identified articles published on similar topics. Finally, we benefited from the helpful assistance of the project heads, inspectors and medical physicians in each of the four countries.

#### 2.1.1. Selection of Countries

The selection of the four countries was influenced by the Southern African TB and Health Systems Strengthening (SATBHSS) project in the four project selected countries supported by the World Bank. A detailed desktop literature review of the published literature was carried out in the African continent to gather data on the organization of occupational health and safety services.

#### 2.1.2. Reviewed Articles Inclusion and Exclusion Criterion

The reviewed literature included technical reports and scientific papers retrieved from article search engines such as PubMed, google scholar and science direct. Computerised search words were “occupational health, occupational health services, primary health care and occupational diseases. The authors then reviewed the obtained articles and technical reports for inclusion in the analysis. The first criterion was to read the article or report abstract to check if it contained the relevant information, while the second part involved reading the entire article or report. A total of 960 articles and 20 technical reports were extracted from the literature search engines. Approximately 603 articles were rejected during the first round of the review, since the information was not relevant to the scope of the project. An additional 293 articles and reports were rejected in the second phase of the review due to non-compliance. Of the 980 searched literature articles, only 80 articles and four reports were used in the final review. In the final review, only eight research articles and three technical reports were used to synthesise the results, since they highlighted OHS in the SADC region. The search words such as occupational health and occupational diseases resulted in a large number of irrelevant literature, which was finally rejected. Other reports and articles were rejected because they were not peer-reviewed or validated before publishing.

#### 2.1.3. Focus Group Discussion

A total of 23 medical practitioners in Zambia were selected according to the districts where mining activities are concentrated. Each district hospital and clinic had one medical practitioner participating in the focus group discussion. A focused group discussion was held with a total of 23 medical practitioners in Zambia serving the mining communities. Semi-structured questionnaires were self-administered to the study participants for completion. The first part of the questionnaires comprised closed-ended questions where the participants were limited to a “Yes or a No” answer, while the second part of the questionnaires comprised open-ended questions where the participants were given an opportunity to express their views.

#### 2.1.4. Requisition of the Occupational Health Database

Thirdly, we reviewed a database on occupational health personnel which included occupational medical practitioners, occupational health nurses, mine health and safety inspectors and occupational hygienists for the four project countries. Information on occupational health services’ arrangement in the four project countries was obtained through a template which required information on the number of occupational health service professionals in each country. The template was completed by the heads of the ministries of health, mines and labour. comprising occupational health nurses (OHNs), occupational medical practitioners (OMPs) labour and mine safety and health inspectors, and occupational hygienists (OHs). Furthermore, the heads of ministries were required to indicate the actual targeted number of occupational health personnel required in addition to what was available.

### 2.2. Data Analysis

Data from the literature review were qualitatively analysed using themes extracted from different research findings. The results from the review were grouped according to themes based on the similarities of findings from different studies conducted in five countries (Lesotho, Malawi, Mozambique, South Africa, and Zambia). Data obtained using questionnaires were exported into a Microsoft Excel 2016 version for further analysis. Incomplete or wrongly completed questionnaires were discarded. Results were tabulated and percentage calculations were based on the absolute 100 value, resulting in the maximum percentage of 100. No statistical analysis was carried out since the study did not seek to compare different situations but only described the allocation of OHS, access and competencies.

## 3. Results

In this section, we have presented the study results in three different parts. The first part provided a review of OHS review in the SADC region. The second part reported the availability of occupational health professionals in four selected project countries, and the third part presented results from a Zambian case study.

### 3.1. Organization of Occupational Health and Safety Services in the Four Countries

In Table 1, several research journal articles and technical reports were reviewed to assess the status of OHS arrangement and provision in SADC region. Almost all the reviewed articles and reports indicated the need to strengthen OHS across the SADC region. Major highlights included increasing institutional capacity in training occupational health professionals, a review of the legislative framework, OHS financing and the provision of OHS in PHC.

### 3.2. Number of Occupational Health Experts (Zambia, Lesotho, Mozambique and Malawi)

In Table 2, a summary of OHS professionals from various ministries across the four project countries is provided. The results are tabulated based on the record as provided by the heads of ministries responsible for occupational health administration in each country. From the results, it is evident that there is an increased shortage of occupational health professionals across the four project countries. It is worth noting that even in countries that reported availability, the professionals were not trained or accredited. Both Lesotho and Mozambique did not have an occupational hygienist or occupational health nurse. The percentages indicate the compliance status for each occupational health professional. Only Lesotho managed to have a 100% allocation of OMPs against the national target. Of concern is the understanding of the ratio of occupational health professional vs. target serving population. Clearly, we have not yet grasped the importance and contribution of OHS in the economy and alleviation of burden of diseases.

Furthermore, the expected number of health professionals at each category suggests that there might be a different understanding regarding the ratio of health professional per population. According to the world health organization, the ratio of 1:1000 population is recommended as best practice. Looking at the projected figures in Table 2, it maybe anticipated that the current allocation are not adequate [24].

### 3.3. Occupational Health Services Provision at Primary Health Level in Zambia

#### 3.3.1. Availability of Occupational Health Services at Primary Health Care Level

Using data extracted from the Zambian questionnaire respondents, it is evident that occupational health services are currently inadequate in the public health facilities/primary health care level. The Table 3 results show that, at the primary health care level, the inception of OHS is limited, with only 11% of the primary health care facilities offering OH services.

#### 3.3.2. Training on Occupational Health Services amongst Zambian Doctors

Among the 23 medical doctors interviewed, (91%), did not have training in occupational health, while only 9% had such training. This finding is a cause of concern as a remarkably high proportion of those practicing are not trained. Lack of training of medical doctors in occupational health may be a contributing factor towards poor diagnosis, pre-placement and health promotion program design.

#### 3.3.3. Risk-Based Prevention Measures Available and Practiced

The results in Table 4 show that pre-placement medical examinations were offered by most institutions. Most of the medical examinations that were done were not risk-based. This is evidenced by the low number of available risk assessment services, which stood at (48%). Exit medical examinations were only available to a limited extent, as only 33% of the respondents indicated the availability of such. The organization of occupational health services in SADC emphasizes the need for health assessment as an essential tool for disease prevention and management. A code of practice requires the employer, after the selection of a suitable candidate, to subject such a potential employee to some pre-placement medicals. The pre-placement medicals are essential in ensuring that the employee is suitable for the assigned activities and attached risks [25]. The standing orders further require that employees, especially those in high-risk zones are subjected to periodic medical assessments every year or at the interval deemed satisfactory according to age and the risk profile [26]. Upon an employee leaving the services of the employer, he/she is required to undergo exit medicals so that a complete medical history of such employee is obtained [27].

#### 3.3.4. Diagnosis Services Available

The results in Table 5 show that most institutions in Zambia do not offer comprehensive OHS, as evidenced by the low proportions of audiometry (24%) and spirometry (14%) services offered. However, a greater proportion of the respondents, 75%, indicated that vision testing was done in most of the institutions. Blood lead measurements were available in 48% of the institutions.

#### 3.3.5. Treatment Services and Training

The results in Table 6 show that primary health care services are the predominant services offered by health institutions, while only 33% offered pneumoconiosis services. TB and HIV services were offered by most health institutions, as evidenced by 95% of the respondents who indicated this.

#### 3.3.6. Health Promotion Services

Most respondents, 86%, indicated that their institutions offered health promotion services, while 76% offered rehabilitation services. In terms of health promotion and rehabilitation, there is a noticeable improvement, which might be aided by the fact that these services are integrated into primary health care services.

## 4. Discussion

From the literature review, only 10 research articles and two technical reports were relevant to the study. This points towards the paucity of published research resources in the field of occupational health in Southern Africa as asserted in the previous study [13]. The literature review findings and results of enquiries with heads of ministries in the four study countries are in concordance with the fact that SADC has gross inadequacies in access to OHS and availability of appropriately qualified occupational health personnel across the medical, nursing, occupational hygiene and safety disciplines. A shortage of trained OHS professionals [10,18,21] and poor OHS legislative frameworks [17,19] are major issues confronting SADC. This standing is further corroborated by the findings of this study, where most Zambian doctors, 91%, lack training in occupational health but are the key providers of health services in primary health care centres. The enquiry from the heads of ministries of health revealed that the four countries are inadequately staffed, and where such positions for occupational health staff were allocated, like in Lesotho, the numbers were way too low for an effective occupational health program at the national level. From the study findings, it is clear that the study countries lack the framework and capacity to deliver comprehensive occupational health and safety issues. Despite the establishment of occupational health service centres in Southern Africa by the Global Fund TB in the Mining Sector project, [13] there still remains a big gap with respect to human resources’ capacity development in the field of occupational health. There is still a need to build capacity that matches the available occupational health facilities in the four SADC countries. The study results show that most Zambian doctors noted that their institutions provided predominantly primary health services, with very few institutions providing extremely limited OHS.

This research reveals the gross inadequacies of OHS provision in the four study countries. It further highlights the lack of integration of OHS into primary health care service centres. A lack of adequate regulatory frameworks in occupational health is the major challenge emerging from the findings of our study. However, the study sample size for the cross-sectional study was from a single country, Zambia, out of the four SADC countries, which might not be representative of the entire populations. The study did not account for variability within countries and persons involved, which might increase the uncertainty of the results. The study is only limited to describing access to OHS and does not include the characterization of institutional capacity, gap assessments and barriers in improving access. Furthermore, the study only focused on four SADC countries, but we do acknowledge that there are many countries in SADC. Only medical practitioners and heads of ministries constituted the sample population. This might present a limitation, especially since nurses, who are often the first contact in health facilities, were excluded. The exclusion of the nurses was based on the funding structure, which only accommodated medical practitioners.

## 5. Conclusions

The findings of our study confirm the inadequacies of access to occupational health services in the four SADC countries. The arrangement of OHS is fragmented and varies in terms of coordination from one country to another. The number of competent practitioners available to render occupational health services remains low, with many countries sitting at less than 15%. For the entire SADC region, the literature suggests that only one country out of the 16 countries has a formal occupational health training program. The lack of training institutions may be among the leading factors of having more untrained professionals in the occupational health facilities. The results obtained in Zambia reaffirmed that a larger proportion of health practitioners (91%) were not trained in occupational health. Most studies indicated that most of the employees or ex-employees are from communities far away from their employment, especially the mining group. The nearest facilities are often community clinics and hospitals, where most of the essential occupational health services are not provided. Therefore, the lack of occupational health centers in most community level emphasizes the urgent need for the provision of OHS at the primary health care level across the four countries under study. It is key to note that poor and near-absent comprehensive occupational health services’ regulatory frameworks could be one of the chief impediments to the development of occupational health services in the region. The study findings conclude that OHS remain greatly constrained in the four study countries with, however, a great potential to expand such services through the transformation of OHS regulatory frameworks and capacity development in the field of occupational health. Since the primary health services offer the first point of access to OHS, it remains a key strategic approach to interface the two services. We strongly recommend building the capacity of occupational health services and integrating them into primary health services backed by comprehensive OHS legislation, which remains the key approach to increased and improved access to OHS across SADC.

## Figures and Tables

**Table 1 ijerph-17-06767-t001:** Results of literature review on occupational health services’ (OHS) legislation and organization.

Author/s	Title	OHS Objective	Key Findings
[10]	Occupational health and safety in the Southern African Development Community	Institutional capacity in offering OHS	Moyo et al. [10] found that OHS in South African Development Community (SADC) lags in many areas, while attention is mainly focused on public health programs such as HIV and TB. They further indicated that most OHS are in the hands of unqualified professionals in the field of occupational medicine. The lack of qualified occupational health practitioners was suggested to be due to the lack of training institutions in the SADC region.
[13]	Review of Occupational Health and Safety Organization in Expanding Economies: The Case of Southern Africa	Occupational Health and Safety (OH & S) regulatory frameworks and service provision challenges.	In the review, the study assessed the arrangements of occupational health and safety (OH & S) governance. It was found that OH & S administration responsibilities were shared amongst different ministries including health, labour, mining and agriculture. The authors concluded that there is high discrepancy in the management of OH & S amongst ministries, leading to systems being fragmented.
[17]	Current Status and the Future of Occupational Safety and Health Legislation in Low- and Middle-Income Countries	Legislations governing OH & S in low and middle income countries.	In this study, it was found out that there is insufficient legislation in the African situation to addresses the need for a comprehensive OH & S. The study also showed that many legislations only focused on chemical hazards, while other important hazard categories were left out. Other legislative considerations such as gender sensitivity and equity were also highlighted.
[11]	Occupational health challenges facing the Department of Health: Protecting employees against tuberculosis and caring for former mineworkers with occupational health disease	Provision of OHS to healthcare workers.	Adams et al. [11] focused on the provision of OHS to the healthcare workers in the Western Cape Province: South Africa. The study found major gaps in the provision of OHS. Health care professionals received very few services, recording the highest score of 83% only for the allocation of a dedicated officer to coordinate the OHS. However, most services such as medical surveillance, biological monitoring and risk assessment achieved a score of below 50%.
[18]	Global Occupational Health: Current Challenges and the Need for Urgent Action	Provision of occupational, funding and institutional capacity.	Shortage of OHS experts across countries as per the survey carried out by the International Commission on Occupational Health (ICOH). Only a third of the surveyed countries had organized OHS for about 50% of their workers.
[19]	Occupational health and safety in the African region: situational analysis and perspectives	Occupational Health and Safety legislation, implementation, and evaluation.	About 37% of countries in Africa had no access to legislations governing OHS. Furthermore, it was found that most of those countries with OH & S legislations did not have sufficient human resources to facilitate the legislative requirements.
[20]	Commentary on the Organisation of Occupational Health and Safety in Southern Africa, the International Labour Organization and Policies in General	Provision of OH & S in SADC region.	In the commentary, it was found that there is a high disregard of OH & S services across the SADC region. Furthermore, more critical OH & S services operate without sufficient competent personnel. Moreover, lack of workplace surveillance is evident and employers disregard employees’ rights to work in safer working spaces. Ncube et al. [20] also reported that employers force employees to work under unconducive working environments.
[6]	Occupational Health Service Delivery in South Africa	Provision of OHS in South Africa.	In the study, it was found that occupational health services for South African employees vary according to the employment categories. Employees in the public sector have a less organized OHS service compared to those in the private sector. The author further emphasized that OHS services in the public health system are hampered by primary health care services. About 94% of the OHS in South Africa were rendered by the private sector, with just a 4% contribution from the government. Moreover, the government enforces legislation protecting workers in the private sector, while they fail to provide the same protection to their employees, despite employees in all sectors being exposed to the same hazards.
[21]	Inequalities in occupational health services for hospital government workers in South Africa.	Organization of OHS in South African health institution.	The study found that about 32% of the health facilities in South Africa had OHS facilities onsite. About 48% of the facilities had a dedicated trained occupational health nurse. Only 9 trained occupational medical health practitioners were employed in South African hospitals.
[22]	Collaborations in occupational health and safety training and skill transfer.	Limitations and gaps in the provision of the occupational health and safety services.	It was reported that most countries in the SADC region experience shortage in critical position within the OH & S space. Moreover, it was emphasized that the shortage can be addressed by collaborative work, where academic institutions support one another in training OH & S professionals.
[23]	Addressing the gaps in OH & S field.	Training of occupational hygiene professionals.	In the project, it was indicated that training the practitioner remains a critical step in advancing the provision of health services in the occupational environment.

**Table 2 ijerph-17-06767-t002:** Number of OH experts per each country against the required number.

Country	OH		OHN		OMP		OS	
	Actual	% of Allocated Position	Actual	% of Budget	Actual	% of Budget	Actual	% of Budget
Zambia	8	40%	2	11%	4	33	0	0
Lesotho	0	0	0	0	2	100%	2	50%
Malawi	5	83%	4	29%	1	25%	0	0
Mozambique	0	0	0	0	1	3%	4	4%

OH = Occupational hygiene, OHN = Occupational hygiene nurse, OMP = Occupational medical practitioner and OS = Occupational safety.

**Table 3 ijerph-17-06767-t003:** PHC services and OHS offered by the health institutions.

Facility Type	Number	OHS Offered	PHC Services Offered
Hospital	17 (74%)	2 (11%)	17 (100%)
Clinic	4 (17%)	0 (0%)	4 (100%)
OHS Center	2 (9%)	2 (100%)	0 (0%)

**Table 4 ijerph-17-06767-t004:** Availability of exposure or disease prevention services.

Risk Assessment	Pre-Placement Medicals	Periodic Medicals	Exit Medicals
10	20	16	7
48%	95%	76%	33%

**Table 5 ijerph-17-06767-t005:** Available diagnostic services at various institutions in Zambia.

Audiometry	Spirometry	Vision Testing	Blood Lead Levels
5	3	16	10
24%	14%	76%	48%

**Table 6 ijerph-17-06767-t006:** Available primary and occupational health services in Zambia.

PHC Services	TB and HIV Services	Pneumoconiosis/Silicosis	Trained on OH
19	20	7	2
90%	95%	33%	10%

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
