# Peer review of "Accessing Occupational Health Services in the Southern African Development Community Region"

_ijerph, 2020, doi:10.3390/ijerph17186767_

Round 1

Reviewer 1 Report

  It consists of three  studies, the first of which is an introduction to the literature, outlining some of the studies on underserved occupational health services in the South African region. The second and third are the actual situation in the area present as tables and figures. However, in the latter 2 studies, it is not clear how one can evaluate that the presented situation is inadequate or not, coparing with other developping area or idial standard area.  However, since the facts they described themselves have certain values and I think it is worth publishing if they rewrite it (especially, studies 2 and 3) in the style of review or opinion.

As minor points,

Acronim "SADC" used without explanation in the abstract and in the text. It must be intoduced in the initial occation as Southern African Development Community (SADC), even if SADC is commoly known abrevation.

Figure 1 has no need to be shown as a figure. The information it contains can be preserved by changing the manuscript as follows,

Among 23 doctors, 21 (91%) did not have training in occupational health while only 2 (9%) had such 188 training. This finding is a cause of concern as a remarkably high proportion of those practicing is not 189 trained.

Also Figure 2 has no need to be a figure, since it is already written in the manuscript. Figure 2 is also, in other way, quite problematic, since the vertical axis started from 70, not from 0. If it is intentional, the authors wanted to show the small difference, as if it was great difference, didn't they?

Author Response

I would like to thank the reviewer of International Journal of Environmental and Public Health – for the detailed and insightful reading of my article document, and the valuable suggestions for its improvement. The comments as presented by the reviewer mainly highlight corrections to be applied to the article regarding use of acronyms, correcting missing text and justification of arguments, as a whole. The reviewer did not point out any flaws in the scope, content, and discussion of presented results and arguments. The reviewer indicated that facts which we have described have certain values and suggest that after minor suggestions are addressed the paper be published. I have considered and responded to the suggested comments as tabulated on the attachment.

Reviewer 2 Report

In this study, the authors made detailed analysis on the access of occupational health services in the southern African development community. Overall, this paper is scientifically sound, but there are still some questions and issues that should be clearly addressed:

  1. In the abstract, a lot of data is mentioned. Are these data credible?
  2. In Section 1, the study hypothesizes that there is limited OHS in primary health care facilities in the SADC region despite high economic growth in the mining sector. If that hypothesis is debatable, future integration may not be necessary. So the hypothesis needs to be justified considering that economic growth may promote investment in occupational health.
  3. In Section 1, this part focuses on describing what other researchers have done. However, it does not highlight what is the research gap in this area and why it should be filled? It is suggested that the authors add some essential contents, i.e. the innovation and significance of this study.
  4. In Section 2, the authors point out the limitations of the study. In order to make the “Material and Methods” part more logical and make the reader be convinced of the results, I suggest move these study limitations part to “Discussion” or “Conclusion” section.
  5. The conclusion part is a summary of the main achievements of the paper. In the conclusion, the creative achievement and main findings should be clearly pointed out. Adding some quantitative analyzed results would be more convincing.

Author Response

I would like to thank the reviewer of International Journal of Environmental and Public Health – for the detailed and insightful reading of my article document, and the valuable suggestions for its improvement. The comments as presented by the reviewer mainly highlight corrections to be applied to the document regarding use of acronyms, correcting missing text and justification of arguments, as a whole. The reviewer did not point out any flaws in the scope, content, and discussion of presented results and arguments. The reviewer indicated that the paper is scientifically sound and made very valuable suggestion to improve the quality of the paper. I have considered and responded to the suggested comments as tabulated.

Reviewer 3 Report

Review of “Accessing occupational health services in the Southern African Development community region”.

The objective of this paper was to characterize the lack of occupational health services in the Southern African Development Community (SADC) region and determine the need of integrating the occupational health services into primary care services in the region. This paper utilized mixed methods including literature review, consultations with ministry heads of four SADC countries, and focus group discussion with medical practitioners in Zambia. The methods are sound, and writings are clear. I have several suggestions below:

Abstract:

  1. Page 1 line 17: Write full name of SADC as it is the first time this term is mentioned in the text.
  2. Page 1 line 19: Specify that 23 doctors from Zambia.

Introduction

  1. Page 2, line 63: Mention Resolution WHA 60.26 by World Health Organization

Methods

  1. Recommend combine section 2.1 and 2.2 and reorganize by specific methods including:
  • Literature review: Suggest including inclusion criteria for each round of article selection.
  • Ministry head consultation
  • Focus group discussion with medical practitioners in Zambia: Recommend adding the interviewing questions in the supplemental materials.
  • Occupational health database

Results:

  1. Are some texts missing at the end of line 157, page 4?
  2. Section 3.3.3 and 3.3.6 both discussed health promotion. What are the differences between the two? Recommend changing the section title to avoid confusion.

I recommend providing additional background information about disease prevention services including pre-placement medicals, periodic medicals, and exit medicals in section 3.3.3. What are the standard occupational disease prevention services in the SADC countries?

Author Response

I would like to thank the reviewer of International Journal of Environmental and Public Health – for the detailed and insightful reading of my article, and the valuable suggestions for its improvement. The comments as presented by the reviewer mainly highlight corrections to be applied to the document regarding use of acronyms, correcting missing text and justification of arguments, as a whole. The reviewer did not point out any flaws in the scope, content, and discussion of presented results and arguments. The reviewer indicated that the methods are sound and well written. I have considered and responded to the suggested comments as tabulated.

Round 2

Reviewer 1 Report

The revised manuscript is sufficiently modified and thought to be publishable in this form.

As a minor point,

in line 16,

"Southern African Development Countries (SADC) countries"

should be

"Southern African Development Community (SADC) countries".

Author Response

Greetings 

Thank you for taking your time in reviewing this manuscript. Your inputs were scientifically sound and was thought provoking during the revision. Once again myself and team we are grateful to your efforts and suggestions.

Best regards,

Daniel
